# The Circular Economy Transformation of Airports: An Alternative Model for Retail Waste Management

Michelle Tjahjono, Enes Ünal * and Trung Hieu Tran

Centre for Design Engineering, Cranfield University, Cranfield, Bedford MK43 0AL, UK
* Correspondence: e.unal@cranfield.ac.uk

**Abstract:** Airport terminals worldwide generate approximately 6 million tons of passenger waste annually. Increased awareness of climate change and global interventions for environmental sustainability requires a reassessment of airports' current methods of waste management. This paper proposes a new design concept solution called circular airport retail waste management (CAWM) for airport terminal retail waste processing, which aims to reduce and ideally eliminate airport waste ending up in landfill or incineration. Given the need for novelty and challenging the status-quo, the double diamond design process was adopted as the research method. The research began by collating the current practices of retail waste processing in airports via a literature review and field observations. Secondly, a critical analysis of the current processes was conducted to identify the intervention points. Thirdly, a concept solution was developed based on the circular economy (CE) 9R framework. Finally, the CAWM concept was delivered to airport waste management personnel for review. CAWM offers a structured way of airport retail waste management practices, including the segregation of nonrecyclable and recyclable waste (i.e., different bin designs, color coding, harmonization of waste colors, improved instructions and signage, various bin locations, training, and installing more liquid disposal and donation stations). Airports can leverage CAWM for greater efficiency and cost-effectiveness regarding airport terminal waste processing, such that more waste can be diverted from incineration and landfill to recovery, which will subsequently help airports achieve net-zero targets. This research contributes to the extant CE literature, especially in the aviation industry context, where the academic discourse surrounding this subject and its peculiarities are limited.

**Keywords:** circular economy; airport retail; retail waste; waste management; zero waste

## 1. Introduction

Retail waste consists of materials that customers and retailers dispose of. Examples of retail waste produced include, but are not limited to, packaging, spoiled food, plastic, damaged fruits, organic waste, glass, assorted metallic waste, clothes, and different fabric waste [1]. Most retail waste is generated by shops, restaurants, and other businesses that form large amounts of packaging and general waste that cannot be recycled and typically end up in landfills.

Airport terminals worldwide generate approximately 6 million tons of passenger waste annually [2], with a high percentage derived from the waste inside the terminal building, often from purchases that are made past security screenings. Unfortunately, waste originating from those purchases has largely been incinerated or simply landfilled [3]. Landfill uses vast amounts of space and can produce harmful gases. Similarly, incineration pollutes the air due to its toxic fumes, which are detrimental to human health.

Excessive airport terminal retail waste may be caused by several reasons. After the security checks, passengers are exposed to various retailers, including restaurants, shops, and eateries, before boarding. These retailers often offer bargains, such as 'three for two', 'buy one get one free', and other common meal deals consisting of a main meal, snack, and drink, which results in unwanted packaging. Due to time pressures before boarding and

the limited volume of luggage to be carried, passengers may feel the need to reduce their baggage load as much as possible by avoiding carrying any unnecessary packaging when travelling, which inevitably promotes less consideration for responsible waste disposal.

Similarly, due to the lack of space and movement that retailers are given in airport terminals, they prioritize occupying as much space as possible in the shops for commercial purposes, such as shelving products, signage, and displays. With less space for waste storage and bins, retailers have no other choice but to use the limited waste disposal options that are available in a small space. For instance, retailers may only have a few bins containing both recyclable and nonrecyclable waste. This may increase the likelihood of contamination of recyclables.

Moreover, the producers of the products sold in airport retail shops package their products in the same way as for regular retail outlets (e.g., in shopping centers/malls) that have no space restrictions (i.e., a large warehouse). With the restricted space, retailers at airport terminals would not be able to deal with the amount of waste and process the packaging waste in the same way that regular retailers do.

It is, therefore, paramount for the stakeholders of airport terminals to better understand the root causes of retail waste and find new initiatives to prevent, reduce, and process retail waste properly. One endeavor to improve their capability to reduce waste is to apply circular economy (CE) principles.

The aim of this paper is to propose a new design concept solution underpinned by the CE 9R framework. The design concept solution, called circular airport retail waste management (CAWM), can be used by airport terminal waste processing facilities to reduce and ideally eliminate waste ending up in landfill or incineration. CAWM offers a structured way for airport retail waste management practices, including the segregation of nonrecyclable/recyclable waste, for instance, the use of different bin designs, color coding, harmonization of waste colors, improved instructions and signage, various bin locations, staff training, education, and installing more liquid disposal and donation stations.

CAWM will contribute to the extant CE literature by providing another dimension to the CE 9R hierarchy, which urges users to consider the relative level of complexity (the time, budget, resources, and effort required to achieve the intervention). This is particularly significant when responding to the current issues surrounding waste management, especially in the area of airport retail waste management, where academic discourse surrounding this subject is limited.

The paper is structured as follows. Section 3 demonstrates the double diamond design process, which has been elected as the research methodology. Section 4 discusses the findings from the research and identifies what needs to be developed in the second diamond phase. Section 5 presents the concept solution called circular airport waste management (CAWM), which is a process that outlines the waste management strategies to reduce and eliminate airport retail waste ending up in landfill or incineration. The paper concludes with Section 6 by summarizing the academic contributions, implications for practice, and suggestions for future work.

## 2. Fundamental Underpinning

One of the common ways airports manage retail waste is by following waste management principles, for example, the 9R framework (Figure 1) and waste hierarchy [4]. The 9R framework (based on the 1979 Lansink's Ladder as the first version of the hierarchy [5] is considered the most comprehensive collection of strategies adhering to the CE principles after the 4R and 6R hierarchies [6]. The model was expanded and modified by the Netherlands Environmental Assessment Agency into nine waste principles from the most preferred method of end-of-product handling (most circular or tight loop) to the least preferred (least circular or linear). "The 9R strategy ensures that materials, products and buildings retain their highest value and stay relevant at the end of their service life" [7]. This hierarchical priority level for product life management is assessed by determining

which strategies adhere most to the CE values and prevent materials from becoming a wasted resource at the end of their service life [8].

| | | | |
|---|---|---|---|
| Circular economy ↑ Increasing circularity Linear economy | Smarter product use and manufacture | **R0 Refuse** | Make product redundant by abandoning its function or by offering the same function with a radically different product |
| | | **R1 Rethink** | Make product use more intensive (e.g. by sharing product) |
| | | **R2 Reduce** | Increase efficiency in product manufacture or use by consuming fewer natural resources and materials |
| | Extend lifespan of product and its parts | **R3 Reuse** | Reuse by another consumer of discarded product which is still in good condition and fulfils its original function |
| | | **R4 Repair** | Repair and maintenance of defective product so it can be used with its original function |
| | | **R5 Refurbish** | Restore an old product and bring it up to date |
| | | **R6 Remanufacture** | Use parts of discarded product or its parts in a new product with the same function |
| | | **R7 Repurpose** | Use discarded product or its parts in a new product with a different function |
| | Useful application of materials | **R8 Recycle** | Process materials to obtain the same (high grade) or lower (low grade) quality |
| | | **R9 Recover** | Incineration of material with energy recovery |

**Figure 1.** 9R Framework (adapted from [9]).

The waste hierarchy [10] is an industry-approved value structure similar to the 9R framework, which is best applied in the decision-making for processing airport waste. Both methods aid the best course of action to eliminate waste going to landfills or incineration [11].

R0: Refuse and waste prevention/avoidance refers to the measures to be implemented at any point during the lifecycle of a product before a substance becomes waste [12]—"from extraction of a raw material to its processing into a functional material to manufacturing of a product, further into packaging, distribution and retail, to the use stage and the end-of-use stage." [11]. Accordingly, "waste prevention policies and specific instruments are devised to address individual stages in the life cycle and targeted at the respective key stakeholders involved." [11].

R2: Reduce and waste reduction consists of decreasing "the quantity generated . . . and to adopt an effective system to manage unavoidable waste" [13]. Moreover, it deals with making optimal and economical use of materials by consuming fewer natural resources and includes "any activity that might contribute to reducing the amounts of waste also decreases transportation emissions and energy necessary to process it [ . . . ] Airports may reuse and repurpose materials by using contractual requirements with tenants to require waste minimization activities such as use of specific materials, cleaners, or paints. The reuse or repurposing of recovered materials also reduces the demand for new materials, for example reducing mining of aluminum ore." [12]. It may also be necessary to consider the waste that needs to be reprocessed before it can be reused [13].

R8: Recycling and waste recycling removes materials from the waste stream and uses the raw materials to obtain the same (high grade) or a lower grade quality and function in a new product(s) [14]. An effective recycling program must be designed with approximately 75% of an airport's waste stream to be recyclable or compostable; for instance, paper stands as the "largest single category of municipal solid waste (MSW) generated" [12]. As a result of recycling, the residual waste quantity decreases, and energy and materials are recaptured. For MSW, airports have the most control over their waste-sorting processes, and implementing sustainable waste strategies can result in benefits, including economic and operational savings. Moreover, staff will need to be trained for these operations, and consideration is given to the placement of special recycling containers throughout the airport as well as procedures for segregation and shipping the recycled materials to their appropriate destinations. For effective and successful recycling, monitoring and

supervision is essential and will require strong leadership skills to co-ordinate the various sectors of the airport to develop a process that is most effective [15].

R9, or waste recovery, is the generation of more useful fuels, electricity, or heat from the conversion of nonrecyclable waste materials that can be used to supply energy. There are several conversion processes, such as combustion (incineration), anaerobic digestion, gasification, and landfill gas recovery. This waste-to-energy process generates a cleaner source of energy by rejecting the need for conventional fossil fuel sources for energy, resulting in lower total carbon emissions [14].

Waste disposal is the least environmentally preferred method of waste processing, but while decisions to reduce, reuse, and recapture materials and energy are currently being utilized, reducing and reusing may not always be possible. Landfills and incinerators are commonly the choices for airport waste that cannot be managed by other processes. However, in some cases, waste-to-energy, as mentioned in waste recovery, attempts to recapture energy through incineration or other processes [12].

## 3. Methodology

The double diamond [16] design process was chosen as the research methodology for the project and the steps to be conducted to achieve each stage of the design process. This method (see Figure 2) contains two diamonds, and the purpose of the diamond shape is to illustrate the two key phases of the design process.

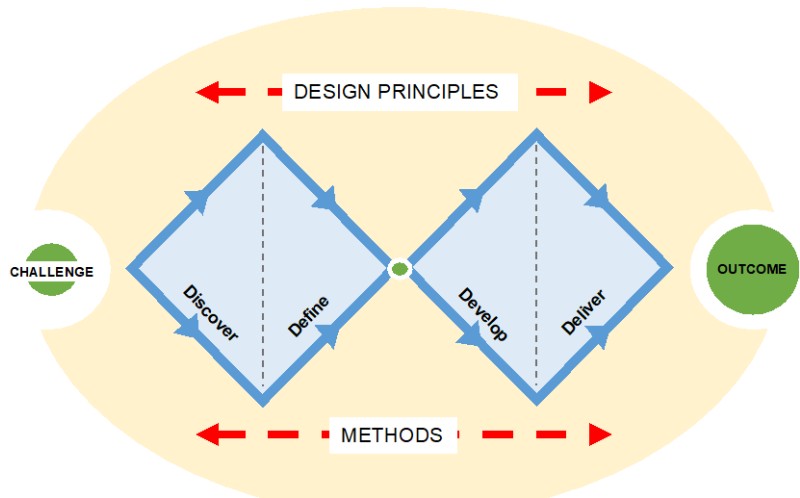

**Figure 2.** Double diamond design process (adapted from [16]).

The first diamond can be viewed as the strategic part of the design process, which builds a foundation of research to aid the direction of the project. In the second diamond, the ideation and exploration of different design concepts take place based on the requirements from the refined scope generated at the end of the first diamond. The second diamond lends itself more to the creative development of the product, service, or system, while the first diamond consists of research and problem identification [17].

### 3.1. Discover

The *Discover* stage contains the divergent (exploring outwards) process, which encourages widely researching and exploring the subject, opening the scope to many possible avenues without limiting the generation of ideas [16]. It was essential in the divergent phase to engage in vast research and allow for the thorough exploration of best practices regarding the CE principles in the context of airports. In-depth literature research was conducted on current airport terminal waste management and CE applications for waste processes and management. The work described in this paper was commissioned by a

major airport in the UK, the management of which was interested in CE applications in airports and who will be referred to as 'the airport' hereafter. Data collection by visiting the airport's Terminal 5 on-site was necessary to conduct observations and interviews with the airport contractors and retailers about the waste journey to identify where there were processes that affected the efficiency and/or effectiveness of the waste processes, as well as the efforts made to minimize waste's destination to landfill or incineration. Photographs were captured for each part of the waste journey, which was an effective way to identify the details and gaps and formulate questions via visual imagery.

### 3.2. Define

When there was sufficient foundational research via a literature review and the scope of the retail waste set, this marked the *Define* stage, which identified the problems and best practices to be synthesized. The convergent phase (refining inwards) takes those broad ideas and narrows down the focus of the findings into defined points. By taking the learnings and findings from the literature review and first-hand observations, these can then be clustered or categorized into best practice principles, which state the key insights and recommendations the concept solution should encompass [18]. Moreover, refining the scope and expressing a list of design requirements was necessary to prioritize the best practice(s) that were identified in the *Discovery* part of the diamond.

### 3.3. Develop

The *Develop* stage involves diverging on various design ideas by experimenting with different ways to illustrate the 9R principles so that the solution can be applied to the target industry. Moreover, it is important to justify why the final concept solution was selected by recording other concept solutions. The final concept solution aims for clarity and cohesiveness (out of all the designs), which can be presented as a deliverable for the airport. The *Develop* stage is not a linear process but is rather iterative, as is shown by the arrows going back and forth in the model. At many points, the design may experience further modifications and can be sent back to the first two diamonds—*Discover* and *Define* to contemplate the changes. The initial problems may need to be reviewed again to gain an overall sense of or a 'bigger picture' of how the concept solution can respond to these problems.

### 3.4. Deliver

In the same diamond, *Deliver* involves a small-scale test of the different solutions, or the chosen solution selected from the *Develop* stage, rejecting those that do not work and making improvements and modifications to the concept solutions that do. The concept solution should be passed onto waste management personnel (via an interview or an onsite visit) as a physical deliverable for the airport to review its viability as a framework for industry use. Since this is a small-scale test, this feedback can be used to make modifications by going back to the initial design(s) and making minor adjustments rather than creating a new concept solution altogether.

## 4. Findings

This section discusses the findings from the research based on the *Discover* and *Define* stages of the double diamond methodology to identify what needs to be developed in the next diamond phase.

### 4.1. Discover–Empirical Data Collection

Three waste producers selected from an extensive list of airport stakeholders (see Table 1) state the typical types of waste generated by each stakeholder that corresponds to tenant waste and/or retail waste.

**Table 1.** Key airport stakeholders and typical types of retail waste generated (adapted from [19]).

| Airport Stakeholder | Types of Retail Waste Generated |
| --- | --- |
| Airport concessionaries and shops | Food, general rubbish, paper, toner cartridges, batteries, lightbulbs, plastic bottles, aluminum cans, packaging |
| Passengers | Food, aluminum cans, plastic bottles, packaging, newspapers, magazines |
| Restaurants | Retail and food and beverage waste, cardboard boxes, paper, plastic items, packaging, food packaging, food wrappers, oils and grease, aluminum cans, plastic bottles, plastic and glass containers |

Several waste-streams that retail waste undergoes arise from both groundside/landside (areas before security screening) and airside (areas after security screening). The groundside areas of the terminal consist of airline ticketing offices, check-in counters, packaging counters, restaurants, cafés, retail outlets, etc. The airside areas of the terminal host the postsecurity check processes, which include duty-free shops, break rooms for airline and airport staff, eateries, and restrooms.

In the *Discover* stage, the waste journey observations were carried out in the airport's Terminal 5, tracking from the terminals to the compactors in the backrooms. The interviews took place with three members of waste management personnel–one external staff member who manages the physical collection of waste and another who processes the waste via the treatment, recycling, recovery, and disposal of waste streams for general waste, as well as the airport's waste manager. At the entrance of Terminal 5, the passengers first encounter a four-bin system with paper cups disposal on the far left (with a white sign), paper and cardboard disposal (with a blue sign), plastic bottles disposal (with a green sign) and on the far right, and general waste disposal (with a black sign). Coffee cups are designated their own bin because they must be separated by lid and cup due to different materials, and thus go through different waste processes. Similarly, sandwich packets must also be separated into their clear plastic case and cardboard constituents. Clear rather than colored bags are used to collect waste as this is the airport's policy for security purposes; however, waste personnel claimed to have future plans for putting a green stripe on the paper and cardboard bin bags (blue) and plastic bottle bin bags (green) to aid in the segregation of waste types throughout the waste journey to the bin room and compactor. There were many instances of liquid leakages, such as with tea or coffee present in the bins, and thus there was clear contamination between wet and dry materials as the passengers disposed of their half-empty beverage cups.

Liquid disposal stations are installed to collect liquids, such as drinks, which passengers empty their bottles into before security screening. This aids in the segregation of wet and dry materials when processing the waste for recycling. However, due to the high costs of installation and current budgets, the airport's waste management were hesitant to install more stations around the airport. The terminal bins are transported to rooms where they are emptied into larger wheelie bins with the same color-coding system as the terminal bins. Additionally, there are grates on the floor as one enters the room where the liquid disposal tank is emptied into. Retail waste is also sent to the bin rooms; however, retailers do not have the same waste sorting system as the airport waste bins due to limited space (i.e., more commercial space) and present fewer incentives to segregate waste at the bin source. Thus, there is less control over the contents of the airport's waste and what waste type reaches the bin rooms.

The last stage of the waste journey within the terminal parameters was compacting, where the wheelie bin contents were dumped into their related compactor. The compactor colors use the same coding system as the terminal wheelie bins, with the exception of paper and cardboard, are colored brown instead of blue. The compactors are collected by recycling facilities, where the waste is transported and processed at the premises. One initiative implemented by the airport is that all retail cardboard, paper, and packaging waste from the airport terminals is sent back for recycling as the lorries for inbound stock are transported

in. This exchange of new and disposed product materials on the same lorry saves on the distance and $CO_2$ emissions that would be produced from two separate journeys.

The airport stressed the importance of segregation at the source due to the costs of segregating material waste at the contractor's processing plant, which charges the airport when more contamination and mixed materials are present, with the extra effort required to separate them. Subsequently, the airport stated a preference for having an onsite recycling facility, where they would be able to quality check the waste streams and, thus, reduce contractor costs overall. Furthermore, the airport stressed the importance of a clean food-waste stream and mentioned their action of recovering oil to then be converted into sustainable aviation fuel.

## 4.2. Define

The problems identified through the literature and the data collection are presented in the *Define* stage, where the problems that exist with waste management are specifically laid out.

Allowing retailers and restaurants to provide single-use plastics, for instance, bottles, serve ware, or plastic bags that the retailers continue to supply and that passengers continue to purchase and dispose of, poses a problem for waste reduction as this waste must sequentially be processed properly. For example, polystyrene and polypropylene plastic usage can potentially leak chemicals into the environment, harming water sources, and the manufacturing of polystyrene generates large amounts of hazardous waste [20].

Due to the continued existence of waste, one of the largest contributors—and thus a defined issue of waste management from the literature review and the data collection—is contamination through the lack of segregation/separation of waste materials at the source. Retailers claim they have neither the time nor available space for a four-bin system at their establishment, so there is little action being taken by retailers to implement an effective segregation method. Moreover, the liquid disposal station remains impractical in segregating waste if passengers continue to dispose of their unemptied cups and bottles in normal bins.

Another case of large amounts of recycling contamination at the airport is the fact that the current system the airport depends on is highly reliant on human behaviors to achieve correct waste management. The airport can only place so much signage, messaging, and color coding, but what ultimately impacts the effectiveness of waste processing, in the end, is the efforts of human manual placement, which is prone to errors. This extends to passengers, who may have their own understanding of waste sorting, which is subject to different factors such as education, background, or even language.

Other issues that have negative impacts on reducing or processing waste at airport terminals are things such as preventing facilities from being shared with other stakeholders and not making it mandatory for airport staff to use reusable coffee cups, flasks, or beverage holders when working in terminals or offices. Moreover, other reasons exist, such as not donating passengers' unused aerosols obtained at security screening and food to local shelters and charities and allowing cargo operations to generate massive amounts of recyclable materials, such as wooden pallets, plastic, cardboard, and paper packaging without utilizing these raw materials for other means. Furthermore, plastic bottles generate large amounts of waste in landfills and this creates harmful toxins produced by incinerating the bottles, but airports that do not utilize technologies to repurpose these plastic bottles will continue to contribute to the harmful effects of these processes.

Finally, a lack of stakeholder engagement through an awareness of their roles and responsibilities and active involvement negatively impacts waste management targets. This is because airports host numerous stakeholders who are involved in multiple activities and operations. The stakeholder map in Figure 3 shows the complexity of the conflicting stakeholder values. Whilst passengers may have strong influence or power for impacting waste generation, as mentioned before, they may have low interest or priority to actively engage in responsible waste disposal. However, the government may be interested in introducing

policies, such as banning single-use plastics in airports, due to their current political party values, thus giving them more power and influence over waste management change.

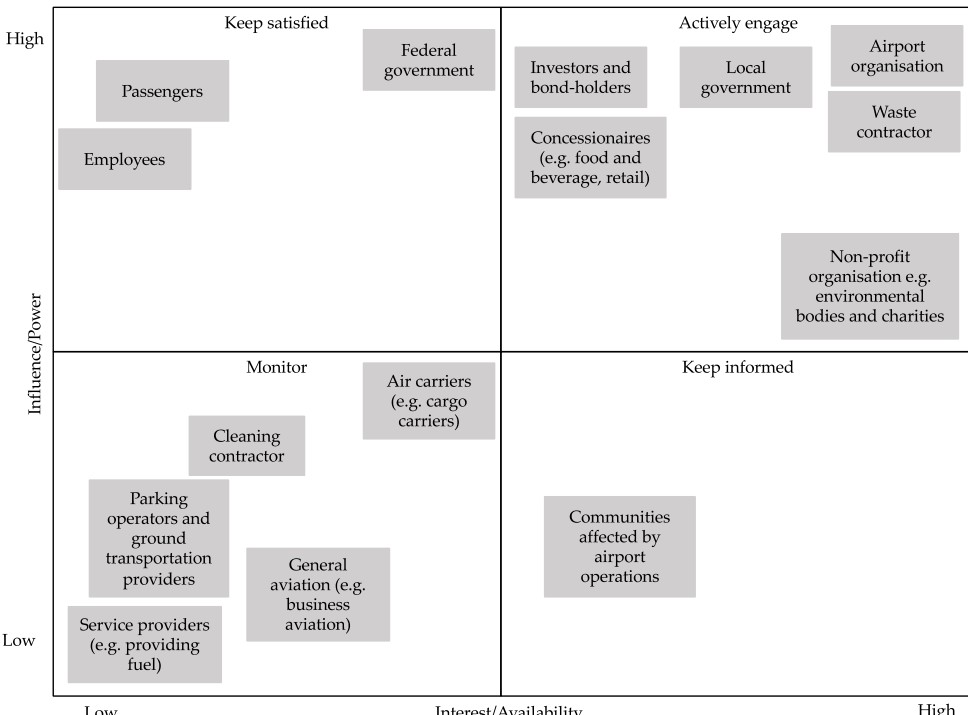

**Figure 3.** Stakeholders for airport terminal retail waste management.

Key behaviors differ based on passengers' cultural, educational, and socioeconomic backgrounds, which informs their attitudes towards consumption and disposal. Furthermore, language barriers pose a hindrance to effectively communicating waste diversion and disposal practices at an airport. The passive participation in waste reduction activities can constitute several factors that hinder the progress of sustainable waste practice. Due to high employee turnover, education through training programs and outreach can only extend so far to help airport tenants comply. A lack of consistency and uniformity throughout the tenant contracts when referring to recycling requirements, adequate training, and effective source segregation present other barriers to sustainable waste management.

### 4.3. Develop

This stage involves analyzing the concepts and best practices to form different design rationales to solve the problems. The concept solution, circular airport retail waste management (CAWM), was influenced by the research surrounding the 9R framework. Rather than providing another top-to-bottom hierarchical framework, such as the 9R framework and the waste management hierarchy, CAWM provides intervention examples for each CE 9R level that can be modified or removed depending on the airport's needs. CAWM also provides another dimension to the hierarchy by presenting a relative level of complexity, which can be used to assess whether the interventions are viable based on the airport's budget, time, and resources and the effort required to achieve the intervention.

Details about the CAWM framework as part of the Develop stage are presented in Section 5—Discussion.

### 4.4. Deliver

The CAWM framework as shown in Figures 4–6 will then be passed onto waste management personnel at the airport for review and feedback. The framework will help the airport waste management department to make applications for the industry to increase

their waste management, which has further, larger implications for diverting waste from landfill and incineration. This will also help the airport achieve net-zero targets and, due to their increased efficiency when applying some initiatives, will consequently have positive economic impacts on the business. Industry waste contractors may also appreciate a new approach rather than the familiar waste hierarchy principles.

# Circular Airport Retail Waste Management Framework

### What is the Circular Airport Retail Waste Management (CAWM)?

It is a framework that shows strategies and examples of waste management applications, their relative complexity level and how they align to the Circular Economy 9R values. This will aid in the decision making for which airport waste management changes are most viable/feasible to apply, which contributes to the overall goal to reduce landfill/incineration usage.

### Who can use it?

Waste management personnel can use it as a 'bigger picture' guiding framework by filling in various examples of airport waste initiatives and prioritising these applications to see which ones are most feasible based on their budget, time etc.

### X-axis: Circular Economy 9R values

The 9R framework by Lansink (2017) presents nine waste principles from most preferred method of end of product handling (most circular or tight loop) to least preferred (least circular or linear).

*Smarter product use and manufacture:*
**Refuse**: Make product redundant by abandoning its function or offer the same function with a radically different product.
**Rethink**: Make product use more intensive (e.g. by sharing product).
**Reduce**: Increase efficiency in product manufacture or use by consuming fewer natural resources.

*Extend lifespan of product and its parts:*
**Reuse**: Reuse product which is still in good condition and fulfils its original function.
**Repair**: Repair and maintain defective product so it can be used with original function.
**Refurbish**: Restore old product and bring it up to date.
**Remanufacture**: Use parts of discarded product in new product with same function.
**Repurpose**: Use discarded product in new product with different function.

*Useful application of materials:*
**Recycle**: Process materials to obtain the same (high grade) or lower (low grade) quality.
**Recover**: Incineration of material with energy recovery.

### Y-axis: Level of complexity

The y-axis measures the complexity level in terms of the investment of time, effort, money and resources needed to implement waste management strategies.

The framework can be used as a template (see right) with boxes to be filled in with various initiatives.

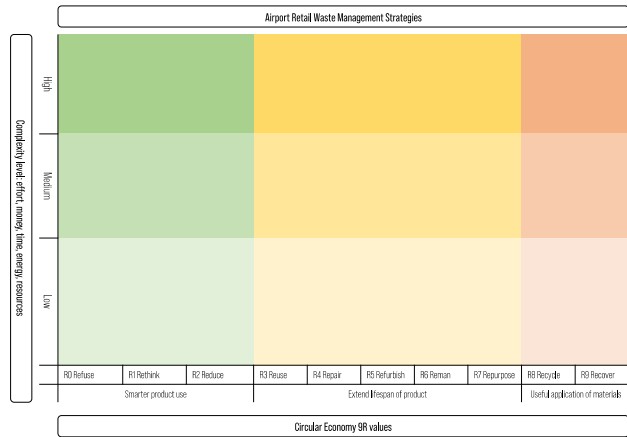

**Figure 4.** Using the circular airport waste management framework.

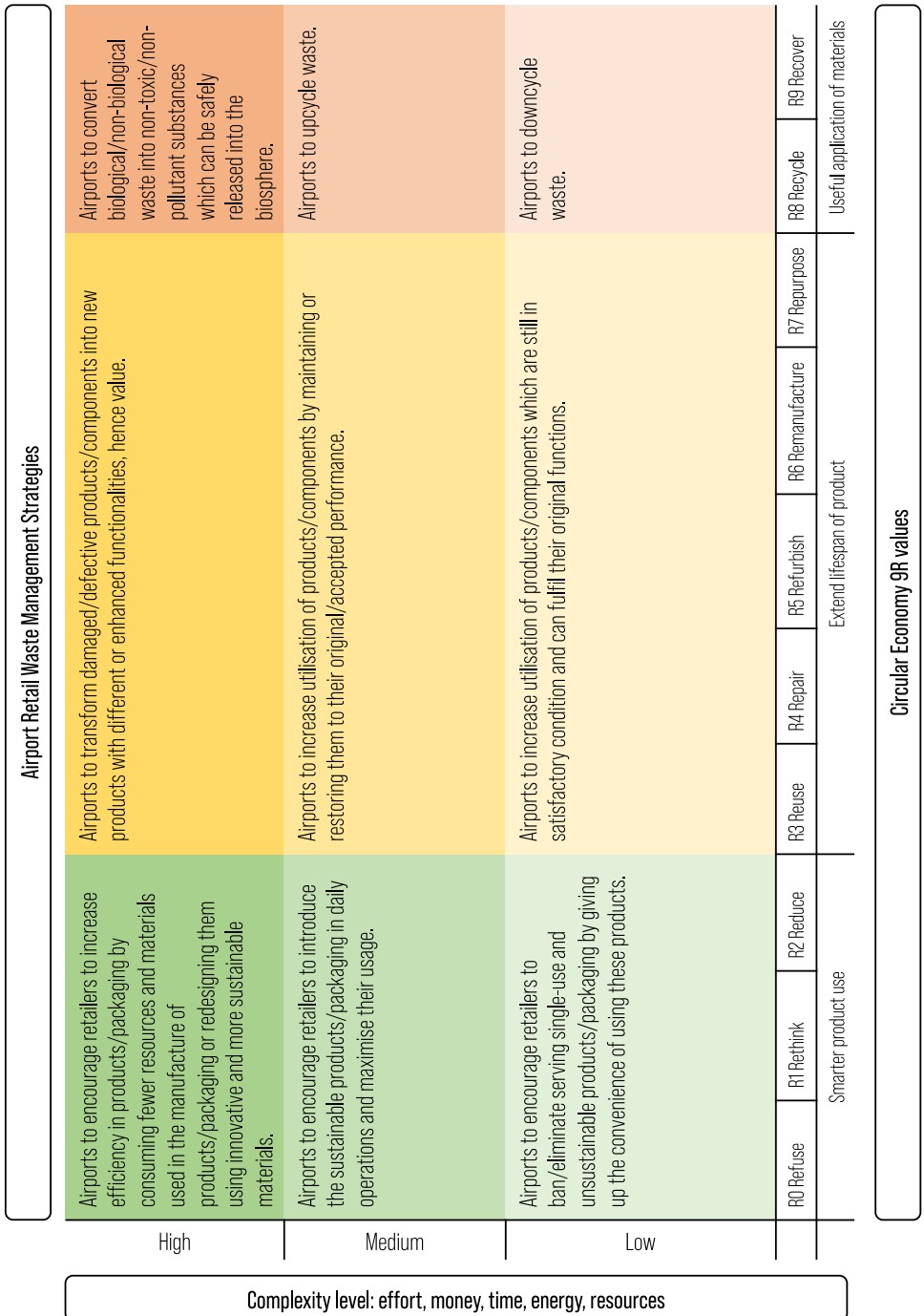

**Figure 5.** CAWM with airport terminal intervention strategies.

**EXAMPLES: Airport Retail Waste Management Strategies**

| | R0 Refuse | R1 Rethink | R2 Reduce | R3 Reuse | R4 Repair | R5 Refurbish | R6 Remanufacture | R7 Repurpose | R8 Recycle | R9 Recover |
|---|---|---|---|---|---|---|---|---|---|---|
| **High** | Use reusable containers for inbound and outbound product stock to and from terminals | *Servitisation:* renting out facilities to others and airport carries out maintenance when necessary | Install more liquid disposal stations to reduce purchase of water bottles | Extract water from liquid stations, separate from other liquids and reuse for watering or washing | Use construction waste to add materials onto equipment/facilities | Retailers use digital signage for correct waste separation messaging | Remanufacture facilities such as waste compactors | Non-recyclable waste can be transformed into airport furniture and renewable fuels | Construction of a mixed plastics recycling facility integrating washing and sorting to recycle rigid plastics | Compostable and disposable plastics converted into oil and then processed into fabrics, low carbon, furniture or jet fuel |
| **Medium** | Collaborate with retailers serving sustainable alternative serve-ware (e.g. biodegradable materials, wood, bamboo compostable stir sticks etc.) | Share facilities with other stakeholders | Cleaning contractor to use refillable/reusable cleaning supplies | Reuse cargo wooden pallets or cardboard packaging by collaborating with shipping companies and new technology | Repair signage by printing new signs | Bins have clear messaging via consistent waste color coding, signage, imagery, icons and instructions | Remanufacture liquid disposal stations | Grass cuttings repurposed into airport furniture | Online system to record waste disposal and transport to make sure correct materials are sent for recycling | Incineration of food waste, meal plates and cups to heat terminal or generate renewable electricity |
| **Low** | Ban polystyrene and polypropylene plastic usage in daily operations | Recycling bins and liquid disposal bins placed near security section of airport | Send back used packaging with inbound stock on same lorry | Mandatory for airport staff to own reusable coffee cups/flasks when working in offices or terminals | Repair bins by repainting messaging or colors | Modify bin signage so that they are displayed at eye level | Remanufacture wheelie bins | Repurposing cardboard into wall fittings | Staff make rounds of collecting waste in bags through terminal to assist in correct waste separation | Food waste, coffee grounds, paper towels, green waste composted |
| | Smarter product use | | | Extend lifespan of product | | | | | Useful application of materials | |

Circular Economy 9R values

Complexity level: effort, money, time, energy, resources

**Figure 6.** Examples of CAWM with airport terminal intervention strategies.

## 5. Discussions

The recommendations described in this section represent the *Develop* stage of the second diamond, where the concept solution, called circular airport waste management (CAWM), has been developed. CAWM contains a graph with the Y-axis (i.e., level of complexity needed to undertake airport intervention) corresponding to the X-axis (each 9R CE strategy). The complexity axis measures the level of investment in the time, effort, money, and/or resources needed to implement waste management changes. The CAWM framework allows connections to be made to the CE via many accounts of best practices demonstrated by several airports that unknowingly adhere to this framework. These airport terminal waste processing and management interventions were also formed by the findings from the airport's Terminal 5 waste journey.

*5.1. Smarter Product Use and Manufacture*

This strategy encourages airport retailers to increase the efficiency of their products/packaging by consuming fewer resources and materials used in the manufacture of products/packaging or redesigning them using innovative and more sustainable materials. Moreover, retailers should introduce sustainable products/packaging in daily operations and maximize their usage.

5.1.1. R0 Refuse

One direct R0 intervention that can be enforced by airports is the elimination or ban of single-use plastics offered by terminal retailers when serving takeaway meals or retail packagings such as plastic utensils, straws, cups, plastic covers, bags, and amenity kits [21]). Airports can decide to work with retailers serving exclusively sustainable alternative serve ware, such as those made of wood or biodegradable materials; for instance, compostable stir sticks [22].

Since single-use plastic water bottles are a highly purchased product, it would be optimal to enforce a ban on the sale of them. Airports should refuse to serve any disposable serve ware, with the restaurants and eateries exclusively offering reusable serve ware (e.g., enduring safe materials), which is washed for reuse rather than disposed of (once passengers have returned them to the establishment).

Furthermore, styrofoam (polystyrene) and polypropylene plastic usage in daily operations should be banned as well. Products transported to the airport terminal can be stored in reusable containers rather than cardboard packaging to eliminate packaging waste completely. Airports can implement 'green concessions' where they reward and thus positively reinforce the responsible waste management of retailers by offering exclusive marketing support from the airport authorities. Takeaway cups can be refused in the airport bins as instructed by the signage or messaging on the bins (e.g., "Do not dispose of coffee cups") for a certain bin. Similar messages can be written on the cups to encourage passengers to return them to the eateries or discard them into the designated coffee cup bins after liquid disposal. These enforcements can be made mandatory by airports for products to adhere to environmental and economic criteria.

5.1.2. R1 Rethink

Airports can share recycling, compacting, composting, and storage facilities with other stakeholders to intensify the usage of that facility rather than each airport individually owning one and not allowing a complex process to be utilized by more than one user. The usage of that facility can be optimized with other shareholders servicing the facility and, therefore, avoiding the production of a one-off facility for each user. The servitization of these facilities means that providing the facility service becomes an ongoing engagement and not a one-off activity [23]. Airports can create a long-term relationship with other stakeholders while renting out their facilities to others and monitoring data for potential maintenance needs. Therefore, maintenance is only required when necessary and avoidable damage can be detected early, which subsequently reduces the overall cost.

5.1.3. R2 Reduce

Installing liquid disposal stations reduces contamination with recyclables when discarding waste, such as beverage cups and water bottles, before airport security screening. It would be even more optimal if the liquids could be separated between water and other liquids so that the water could be directly reused for other purposes, such as watering or washing.

Subsequently, airport terminals can install water fountains or refill stations on the airside of security and around the airports to reduce the purchase of single-use plastic water bottles. Airports should advocate for the use of refillable containers for cleaning supplies and other fluids when maintaining the hygienic upkeep of airports. This way, they can avoid the constant disposal of things like cleaning wipes (especially wipes that do

not break down in water) and avoid harmful cleaning fluids that affect the local ecosystem when disposed of. When disposed of, the packaging from bulk products should be compacted to as small a size as possible, although this only reduces the volume of packaging in cargo transport services. Airport terminals should target eateries, such as Costa, Starbucks, Pret a Manger, and Caffè Nero [24], where takeaway materials are most abundant as the purchased products produce the largest quantities of contaminated materials for preprocessing; therefore, the instructions for their correct disposal should be mandatory for retailers—not solely the passengers or the airport's responsibility. Therefore, to reduce contamination, there should be more areas within the terminal to pour liquids away, with each recycling and nonrecycling bin being accompanied by a liquid disposal bin to encourage the separation of dry and wet materials. Since no liquids over 100 mL can pass through security, authorities should optimize the recovery of uncontaminated high-grade plastic bottles, especially if there is a security presence near the bins [25]. Overall, passengers can be encouraged to carry fewer liquids, gels, and aerosols as carry-on items through various messaging or signage. Other interventions include encouraging delivery partners to communicate the message about reducing the packing materials to their owners.

### 5.2. Extend Lifespan of Product

The second strategy encourages airports to increase the utilization of products/components that are still in satisfactory condition and can fulfill their original functions. Furthermore, they should continue utilizing products/components by maintaining or restoring them to their original/accepted performance. Airports can also transform damaged/defective products/components into new products with different or enhanced functionalities, hence increasing their value.

### 5.2.1. R3 Reuse

Unconsumed food from various eateries in the airport vicinity and other common airport discarded products, such as new aerosols, can be reused via donations to local shelters. Furthermore, airport staff should own reusable coffee cups when working in terminals or offices to reduce the waste generation from workers. Water disposed of before security screening can be extracted from central collection points and reused for watering or washing. Moreover, terminal cafés can offer takeaway reusable cups that can be returned to them after use [24].

This can be introduced into the terms of contract agreements with the stakeholders to achieve sustainability and zero waste targets. Cleaning companies should use green cleaning supplies such as reusable wipes and containers rather than repurchasing newly packaged cleaning fluids. If there were interventions for cargo handling and shipping companies to reuse wooden pallets or reduce the paper and cardboard packaging, this would significantly decrease the excess waste generated.

### 5.2.2. R4 Repair

Airports can repair facilities and equipment rather than purchase new ones. For instance, repairing furniture by using materials that are disposed of from other waste (e.g., construction waste). Airports should consider repairing their bins rather than purchasing brand-new bins. If there is damaged signage or patchy paint that can be repaired via repainting or printing a new laminated sign, this would be much more beneficial than disposing of the whole wheelie bin.

### 5.2.3. R5 Refurbish

The recycling bin system should be modified, so bins have clearer messaging via familiar waste colors, signage, and instructions for better waste segregation at the source.

Better sorting, segregation, and separation can be implemented through a refurbished four-bin system. A color-coding system and the harmonization of these colors throughout all the bins are essential as it adds more visual aid to the signage and allows for more clarity

of correct bin disposal. This is more crucial in the context of an airport where there may be language barriers as non-UK passengers' understanding of recycling may differ from their homeland but also from airport to airport.

Therefore, it is crucial to clearly label (with appropriate iconography) and indicate which materials are to be recycled through which bin to increase the recycling rate and reduce the contamination of the recycling bins. The labels on the bins might currently be displayed at the waist level or below; if these labels are not seen at eye level, then they are less likely to be read. The convenience and location were also stressed as many tenants tend to put away their rubbish in the bin with the closest proximity regardless of whether it was the correct bin or not, which risks contamination of recyclable materials [24]. Thus, placing the bins next to cafés, restaurants, or centralized food court sorting stations increases the ease of disposal and may be more effective for tenants to separately dispose of other materials, such as paper cups or plastic bottles. For example, recycling plastic bottles would be more effective if placed near the security section of the airport.

### 5.2.4. R6 Remanufacture

Airports should consider if there is any equipment, such as waste compactors and collectors, that can be disassembled and then remanufactured. The components for the wheelie bin should be easily removable, and the spare parts can be purchased from various websites. For instance, if a wheel on a wheelie bin was faulty, rather than purchasing one new wheelie bin, it would be more beneficial to purchase a new wheel and swap the faulty one for the new one and only make repairs when necessary, thus optimizing the usage of that wheelie bin.

### 5.2.5. R7 Repurpose

Plastic bottles can be repurposed into fabrics for clothing postprocessing. Recycled polyester can be converted from recycled plastic bottles by being shredded into flakes by a machine that runs on renewable energy. The flakes are melted down into pellets and are extruded into yarn, which is then knitted, cut, and sewn into clothing [26]. Nonrecyclable waste can be transformed into airport furniture to avoid purchasing new furniture, and renewable fuels can also be formed from these nonrecyclables.

### 5.3. Useful Application of Materials

Airports can convert biological/nonbiological waste into nontoxic/nonpollutant substances that can be safely released into the biosphere. For processing waste to obtain the same or lower-grade material, airports can both upcycle or downcycle waste.

### 5.3.1. R8 Recycle

By working closely with vendors and retailers, airports should manage waste more effectively via the correct separation, reuse, and minimizing of waste while encouraging passengers to consume and dispose of waste responsibly.

One way to pressure retailers to be waste responsible is by enforcing the costs of the waste disposal back to the waste producers via "originator pays the price"[15], whereby the retailers are taxed on how much of their waste is improperly disposed of, or the amount of packaging used which is nonrecyclable. Materials that should be collected for recycling are mixed waste papers, cardboard, foils, mixed glass, wood, and metals while avoiding wet or other substances that could potentially contaminate the effectiveness of the recycling process. An online system to record waste disposal, transport, and processing would be useful for documenting the disposal chain to allow for quality checks or, in this case, to monitor the correct materials passing through to recycling facilities. However, recycling facilities should be constructed within the terminal or airport to reduce vehicle movements from the airport to the facility to enhance diversion, and similarly, sorting, composting, food digesting, and bottle-compacting facilities should be available within the groundside of the terminals.

### 5.3.2. R9 Recover

The energy acquired from the incineration of food waste, plastic meal plates, and cups can be recovered to heat part of the terminal or generate renewable electricity for other purposes. Plastics can be pyrolyzed (by avoiding potential externalities with necessary measures), which is a technique that uses intense heat and an oxygen-free environment to break down plastic waste into simpler compounds. As a result, fuel oil—sometimes referred to as pyrolysis oil—is produced and can be used as a source of energy. Through the pyrolysis process, plastic waste is converted into valuable products while reducing the environmental impact of disposing of plastic waste.

Anaerobic digestion converts organic waste, such as the food waste from eateries, into biogas/biodiesel (secondary), and similarly, food waste, coffee grounds, paper towels, and green waste can be composted (ideally) at a facility within the airports.

Composting organics, such as paper towels from washrooms, can be used as a filler material for the soil [15]. If there were a composting facility on the groundside of the terminals, local farmers could purchase the fertilizer; subsequently, airports could purchase the produce from these local farmers; the remains of the produce would then be disposed of in the bins and, in turn, would be composted again, thus closing the loop.

## 6. Conclusions

Human behavior has been identified by the airport as one of the most common causes of waste separation and segregation problems. For passengers, waste handling is such a low priority during their traveling procedure that there is much room for error via misplacement which, if added up, contributes to contamination, leading to more waste that could have been recycled. This is problematic as the contaminated waste is then classified as general waste, which ends up being sent to landfil or incineration. There were periods of time when the staff at the airport underwent job rotations (employees on annual leave or staff changes), and the result was a significant increase in the lack of segregation/separation of waste, which shows that those who are not familiar (perhaps due to lack of training) with the process (1) do not know the procedure, (2) know the procedure but do not care (self-interest), or (3) know the procedure but do not know the wider implications of the errors within the waste journey that would lead to more waste being sent to landfill/incineration and, therefore, the importance of separating/segregating at the source.

The CAWM framework provides different options for airports through various intervention examples and the 9R values they correspond to that can improve their environmental impact from waste generation. However, the level of complexity allows airports to map which initiatives are feasible and viable at the time based on their budget, effort, resources, and available energy since these factors are relative to each airport's circumstances. CAWM may be most useful for the stakeholders, such as airport terminal waste management personnel and waste contractors, to be implemented into their airport terminal waste methods. Thus, they have the most power and influence to make these initiatives a reality (as shown in Figure 3).

### 6.1. Academic Contributions

The proposed concept solution will contribute to the extant CE literature, especially for airport retail waste management, where academic discourse surrounding this subject is limited. Due to the limited discourse, this may lead to less peer review from independent researchers in this area of CE research for airport retail waste, which has a significant environmental impact. This concept solution was highly influenced by the 9R framework [5] that supports the CE principles, especially in providing options for R0 and prevention, which accounts for the technical, economic, and political scarcity of raw materials. One of the additional benefits of preserving resources is the usage of less energy, which supports contributions towards climate change solutions. It is clear that, nowadays, designs for recycling cannot be avoided when developing products, equipment, or infrastructure, but airport retailers must take a further step to prevent excess, unnecessary waste from even

forming in the first place. CAWM will enable more CE discourse, which may increase the validity and significance of the research that subsequently determines whether this research can be disseminated further in the public domain. The airport authorities, government, and retailers would be more inclined to believe that CE research has substance and application in the industry so that there might be more movement in changing waste management policy. In European legislation, legal means have been put in place to accomplish the higher levels of the waste hierarchy and the 9R framework in line with the chain management of the CE. Though the airport was the focus of the investigation to obtain data on the waste journey and the current initiatives airports use in accordance with the 9R framework values, the CAWM framework is sufficiently generic to be utilized as a template framework by airports in general.

### 6.2. Implications for Practice

The CAWM concept solution offers better practices of waste management. The added element of the complexity levels aids airport waste management personnel to better understand their situation based on whether it is feasible or viable for them at that time. This may lead to greater efficiency and cost-effectiveness in airport terminal waste processing; more waste can be diverted from incineration and landfill, which will subsequently help airports achieve net-zero targets.

### 6.3. Limitations and Future Work

Some limitations of the CAWM framework may be that airports need assistance with the templates in utilizing the resource, which is filled in with examples of initiatives. Starting from scratch might be challenging and time-intensive for airports in generating their own initiatives, or airports may not know what initiatives correspond to what 9R values and principles based on the CE research. Thus, the deliverable may need to be a complete filled-in framework with initiative examples that are suitable for airport waste management so that airports can modify them and adapt them according to their time, resources, budget, or available effort.

At the time of our research, the airport was understaffed, and this may have led to the results being data-limited, and therefore, detailed and supported recommendations cannot be made specifically for the airport, but they may be more suited to airport terminals in general. Future work may focus on a more detailed empirical validation of the CAWM framework at the airport and in other airports as a case study work.

**Author Contributions:** Conceptualization, M.T. and E.Ü.; methodology, M.T. and E.Ü.; validation, E.Ü., T.H.T. and M.T.; formal analysis, M.T.; investigation, M.T.; writing—original draft preparation, M.T.; writing—review and editing, E.Ü. and T.H.T.; visualization, M.T.; supervision, E.Ü. and T.H.T. All authors have read and agreed to the published version of the manuscript.

**Funding:** This research received no external funding.

**Institutional Review Board Statement:** Not applicable.

**Informed Consent Statement:** Not applicable.

**Data Availability Statement:** The data presented in this study are available on request from the corresponding author.

**Conflicts of Interest:** The authors declare no conflict of interest.

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
