# Peer review of "The Circular Economy Transformation of Airports: An Alternative Model for Retail Waste Management"

_sustainability, doi:10.3390/su15043860_

Round 1
Reviewer 1 Report
The manuscript titled “Circular evonomy transformation of airports: An alternative model on retail waste management” describes a model that potentially contributes to the reduction of airport waste and to the promotion of a circular economy, which is important to reduce problems and to increase awareness of the problems. The manuscript deserves to be published after minor corrections.
L37, remove ‘)’.
L39, make referencing style consistent.
L260, make 2 in subscript.
L274, revise ‘be they bottles’
Figure 3, although readers will understand the map eventually, it is not quite well organized. Font and box sizes are different in each box. Make it consistent and add grid or anything that can help understand.
L521-528, the authors used the word ‘oil’. Plastics can be converted into what ‘oil’? Does this mean fuels (hydrocarbons) and syngas? When plastics are pyrolyzed or gasified, carbons, short- and medium-chain hydrocarbons, or syngas can be recovered. The authors also used ‘oil’ from kitchen food waste. This word cannot be used in totally different areas. Revise this section.
Author Response
Reviewer #1 |
|
|
|
Comment/Suggestion |
Response |
|
|
The manuscript titled “Circular economy transformation of airports: An alternative model on retail waste management” describes a model that potentially contributes to the reduction of airport waste and to the promotion of a circular economy, which is important to reduce problems and to increase awareness of the problems. The manuscript deserves to be published after minor corrections.
|
Thank you. |
L37, remove ‘)’. L39, make referencing style consistent. L260, make 2 in subscript. L274, revise ‘be they bottles’
|
These have been revised. |
Figure 3, although readers will understand the map eventually, it is not quite well organized. Font and box sizes are different in each box. Make it consistent and add grid or anything that can help understand.
|
This has been revised. |
L521-528, the authors used the word ‘oil’. Plastics can be converted into what ‘oil’? Does this mean fuels (hydrocarbons) and syngas? When plastics are pyrolyzed or gasified, carbons, short- and medium-chain hydrocarbons, or syngas can be recovered. The authors also used ‘oil’ from kitchen food waste. This word cannot be used in totally different areas. Revise this section.
|
These have been revised. |
Reviewer 2 Report
The presentation reflects the present state of knowledge. The paper is very well structured. The Introduction section is good, in this section the authors present clearly the objectives and the main contributions of the study. The authors provided sufficient background and include relevant references. The method is adequately described. The results are clearly presented. The conclusions are supported by the results. Authors should follow the format of the reference.
Author Response
Reviewer #2 |
|
|
|
Comment/Suggestion |
Response |
|
|
The presentation reflects the present state of knowledge. The paper is very well structured. The Introduction section is good, in this section the authors present clearly the objectives and the main contributions of the study. The authors provided sufficient background and include relevant references. The method is adequately described. The results are clearly presented. The conclusions are supported by the results. Authors should follow the format of the reference.
|
Thank you. The reference list has been revised. |
Reviewer 3 Report
I like the article and and I consider it partly original.
The submitted article is well written and clearly readable.
The article presented contains an adequate amount of literature used. The current scientific literature is used in the article.
I have only these comments:
a) In the present article would be appropriate to clarify how experimental identifications are used in practice.
b) The article would be appropriate to explicitly indicate the scientific benefits of the proposed solutions.
c) The mathematical apparatus could have been applied to a greater extent.
d) The images have poorer graphic quality and readability of values and it would be appropriate to improve them.
It is clear from the presented contribution that it will be necessary to continue the research.
The paper is well structured. Conclusions are clear, in line with the main text. The manuscript is clear written and could be interesting for the researchers from this field.
Author Response
Reviewer #3 |
|
|
|
Comment/Suggestion |
Response |
|
|
I like the article and I have only these comments: The article presented contains an adequate amount of literature used. The current scientific literature is used in the article. In the present article would be appropriate to clarify how experimental identifications are used in practice. It is clear from the presented contribution that it will be necessary to continue the research. The paper is well structured. Conclusions are clear, in line with the main text. |
Thank you. |
The article would be appropriate to explicitly indicate the scientific benefits of the proposed solutions.
|
This has been added on Lines 78-83. |
The mathematical apparatus could have been applied to a greater extent |
Thank you. |
The images have poorer graphic quality and readability of values and it would be appropriate to improve them |
We have tried to improve the quality of images. |